# Aquaculture Sustainability Assessed by Emergy Synthesis: The Importance of Water Accounting

Úrsula da Silva Morales [1], Marco Aurélio Rotta [2], Darci Carlos Fornari [1] and Danilo Pedro Streit, Jr. [1,*]

1   Aquam Research Group, Animal Science Research Program, Federal University of Rio Grande do Sul—UFRGS, Bento Gonçalves Avenue, 7712, Porto Alegre 91540-000, RS, Brazil
2   Department of Agricultural Diagnosis and Research, Secretary of State for Agriculture, Livestock and Rural Development—SEAPDR, Porto Alegre 90150-004, RS, Brazil
*   Correspondence: danilo.streit@ufrgs.br; Tel.: +55-096-991378724

**Abstract:** Aquaculture is one of the protein production activities with the most significant potential for global development. It is one of the fastest growing in recent years, mainly because of its efficiency in transforming feed into meat. However, the increase in aquaculture production raises some concerns, especially regarding the proper use of natural resources such as water, which is fundamental in aquaculture production systems. On the other hand, numerous systems, models, and production techniques have been developed and used to manage resources and reduce the negative impacts of the activity. However, it is not known which production systems and management practices are more sustainable, although the development and application of these technologies are crucial and profoundly influence this aspect of production. Emergy is a method that considers the contribution of nature and economy in the creation of the product and service, excluding the strictly monetary character present in conventional economic evaluations, being a model used to measure the level of sustainability in productive systems. In this sense, this study characterized the use of emergy analysis in aquaculture systems and discussed the main applications and potential uses, in addition to identifying the importance of water in the production and better destination of this resource for the economic and sustainable development of aquaculture. The systematic review methodology identified 17 articles using emergy analysis to assess environmental, economic, and social sustainability. The production systems evaluated vary between monocultures and polycultures at different production levels (intensive, semi-intensive, extensive). When all these particularities are transformed into the same unit (emjoule or solar joule), it is possible to compare different scenarios. As a primary resource of nature, water deserves more attention in the emergy accounting of aquaculture systems. It was shown the importance of a more detailed water analysis considering its effective use, impact (alteration or variation in its quality), and flow for a correct emergy analysis as a tool to promote the maintenance of the aquaculture activity over the years, which has in the water its most significant wealth.

**Keywords:** sustainable production; sustainable accounting; fish farming

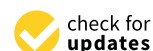

## 1. Introduction

According to FAO [1], the consumption of aquatic foods has increased in recent years, driven by population growth and an increased preference for healthy animal protein sources. In addition, with technological advances, aquaculture has excelled in producing aquatic organisms based on the introduction of new production techniques, affordable costs, and significant gains in productivity and quality animal protein production [2]. On the other hand, the increase in aquaculture production results in numerous concerns about the future of the activity, especially regarding the negative environmental impact it can cause [3], especially regarding the proper use of natural resources, such as water, fundamental in aquaculture production systems [4].

According to the production system adopted for the cultivation of aquatic organisms, natural resources can be used irresponsibly, which interferes with the maintenance of biodiversity, either through the eutrophication of rivers or the impacts of inadequate production practices [5]. On the other hand, aquaculture can positively affect the environment, such as ecosystem services and the production of effluents for agricultural irrigation [6].

With the market potential and the increase in the number of producers, it is necessary to adopt management to maintain the activity. One of the alternatives is the adoption of sustainability precepts [7]. From this perspective, issues related to environmental impacts, economic viability, social equity, and arrangements constitute the sector's governance [8]. However, researchers worldwide are challenged to evaluate and compare the different aquatic production systems of organisms within the context of sustainability.

The numerous methods, models, and techniques to measure sustainability in production have been developed and used to manage resources, reduce the negative impacts of productive activity and make aquaculture more sustainable [9]. However, it is unknown which of these methods is suitable, revealing an accurate portrayal of the activity.

Sustainability assessment methodologies can be applied to show the weaknesses and strengths of each production system and indicate strategies to improve them [10]. Among these methodologies, the emerging synthesis can provide factual information for decision-making and guide sustainability [11,12], standing out for being a flexible and scientifically robust method [11].

Furthermore, this method can provide factual information for decision-making and guide sustainability [12]. Thus, a holistic view of the applicability of emergy assessment in aquaculture systems is necessary to guide future studies and propose alternatives for the sustainable use of natural resources and economic development of the activity. Thus, the underlying question in this review is: "What are the contributions of emergy analysis and the impact of water emergy on aquaculture production?" It intends to characterize the use of emergy synthesis in aquaculture systems and discuss the main applications and potential use, in addition to identifying the importance of water in the production and better allocation of this resource for economic and sustainable development in aquaculture.

## 2. Materials and Methods

### 2.1. About Data

Initially, a search for data was carried out in different electronic databases to support the systematic review and was based on a structured search developed through digital databases (Figure 1). Several search keys were tested based on adaptations of the PICo strategy ('Population,' 'Interest,' and 'Context') using a series of alternative terms in the characterisation of each term. However, as it is a limited area of research, it was decided to use a simple key to reduce the chance of exclusion from any studies.

Thus, the search words "aquaculture" (population) and "emergy" (interest) were used to find a significant number of studies, even if this represented a large number of initial results that would need to be critically evaluated in the selection steps. The search was conducted in the main electronic databases available for research (Web of Science, Scopus, and Science Direct) from August to October 2020. However, as it is a methodology of increasing use in aquaculture, the research period was limited to 2010–2019.

The search results in each database were exported to reference management software Zotero (version 5.0.96). Duplicate references were identified and deleted. The studies were then critically evaluated for their adherence to the research question. Initially, the titles and abstracts of each publication were evaluated, followed by a complete review of the published work. Finally, works that applied the methodology of emergy assessment in aquaculture systems (selection criteria) were selected.

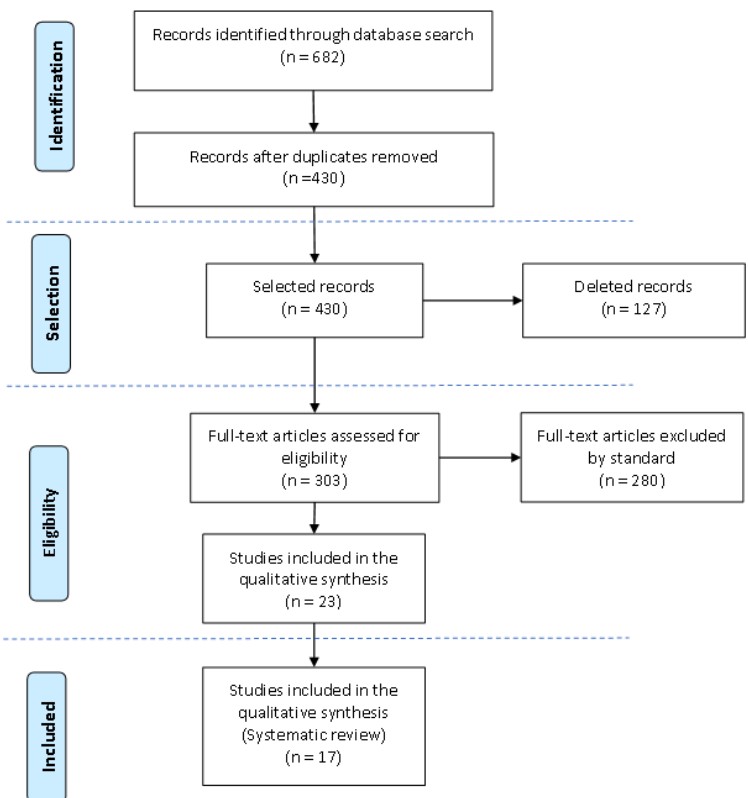

**Figure 1.** Analysis flowchart used in the development of the systematic review.

The selection and evaluation of the eligibility of each study were carried out by two reviewers independently. A record was only removed from the database when there was a mutual agreement, and a third reviewer was consulted in case of disagreement. No limitations of geographical origin were applied in the research or selection stages.

A backward search strategy was used to ensure that the largest possible number of studies was obtained. In this step, the reference lists for each selected article were checked using the same selection criteria as previously described. Finally, the selected articles were evaluated for their quality and the relevancy and information for the description of the theoretical model proposed by each study and then were transferred to electronic spreadsheets. After descriptive evaluations, comparisons between studies were made.

To illustrate the systematic review were used two qualitative analysis software. Infogrann, which makes it possible to create the word cloud with the frequency of use, and Ucinet—version 6.747, which makes it possible to build a network of authors and co-authors and boost the interaction of published articles.

### 2.2. Emergy Accounting

Emergy accounting can be defined as the availability of energy used in direct or indirect transformations to produce a product or service [11]. The emergy, or energetic memory, allows the survey of all factors that contribute to the production of goods and services in the same denominator: the energy of solar radiation equivalent or necessary for the integral production process that has emjoule or seJ as a measure [12].

Emergy accounting is based on thermodynamics principles, systems theory, and systems ecology [13]. Thus, flows of resources not exchanged in the market, including solar radiation, wind, and waves, can be internalised in economic production and valued by emergy [14].

These flows also incorporate matter, energy, money, and the necessary work, culture, and information, which can be aggregated in this methodology to account for their respective contributions to production processes [12]. The emergy analysis unifies nature's

resources and the economy in the same measure, revealing the vast and branched energy chain that unites the system's parts [15].

For example, the sun, fuel, electricity, and human services can be placed on a common basis, expressing them all in solar energy emjoules [16]. Therefore, the emergy analysis quantifies and valuations of their contributions to renewable and non-renewable energy sources, which other techniques generally do not count or only partially count [17].

Renewable resources (R) are extracted from the environment and have the capacity for temporal and spatial renewal faster than their consumption (solar energy, wind energy, rain energy, etc.). Non-renewable resources (N) are stored in nature; however, their consumption is faster than their renewal capacity (coal, oil, fresh water, etc.). The resources from the economy (F) refer to materials and services from other regions outside the studied system's borders [18].

The systems of nature and humanity are part of a universal energy hierarchy and an energy transformation network that unites all systems [19]. As it is a method that accounts for the energy expended in the production processes, it is crucial to recognise the quality and functionality of each type of energy used for the generation of resources given the generation of the energy hierarchy. Therefore, energy transformations can be arranged in an ordered series to form this energy hierarchy [20].

Based on this understanding, the transformity of a product has been calculated by adding all emergy inputs of the process and dividing it by the energy from the product [21]. It is essential to highlight that carrying out the analysis of emergy flows from the systems is necessary to have information about the transformity (addressed as solar energy joules (seJ) per joule of energy).

### 2.3. Emergy Analysis Proceedings

The emergy accounting process follows some linear steps proposed [11]: (i) survey of the history of the place of study; (ii) drawing up a diagram; (iii) setting up the emergy assessment table; (iv) calculation of the emergy indices and (v) interpretation of the results. Finally, the analysis makes a systemic diagram of the studied object [22] (Figure 2).

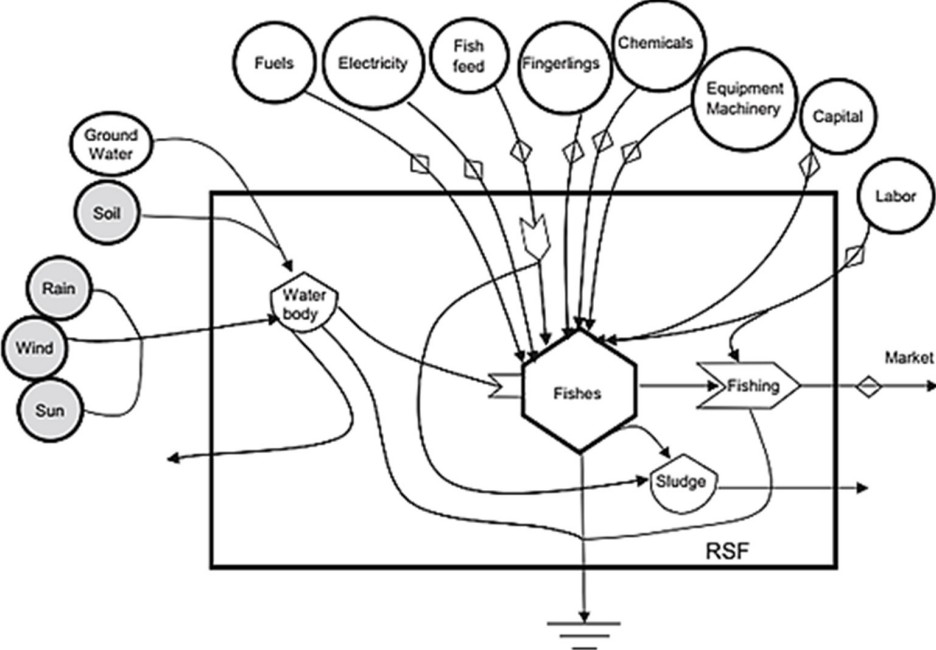

**Figure 2.** Example of an energy system diagram of fish production in a recirculation system. Source: Wilfart et al. (2013) [10].

This diagram is composed of specific symbols of emergy accounting representing the production process (Table 1) and identifies all the resources of the studied system and their interactions and outputs (Table 2). The emergy flow evaluation results infer the emergy analysis from the calculated emergy indices. For this study, the results of the indices for aquaculture will be discussed.

**Table 1.** Main symbols used in constructing diagrams for emergy assessment proposed by Daley (2013) [22].

| Symbols | Name | Description |
|---|---|---|
| | Flow | It determines the flow of energy, information, organisms, materials, etc. |
| | Source | Represents an energy source that supplies power or flow. |
| | Stock | It represents an Energy reserve within the system, a compartment of energy, material, information, etc. |
| | Power sink | Represents the energy degraded during a process, leaving the system as low-intensity energy. According to Thermodynamics' second law, the dispersed energy can no longer perform work (loses its usefulness). |
| | Interaction | It represents a process that combines different types of energy and materials to produce a different action or resource, a transformation that uses two or more flows of different stocks necessary for a production process, producing a new resource. |

**Table 2.** Emergy indices need to be calculated for the agricultural production system, according to Odum (1996) [11].

| Indicators | Description | Equation | Definition |
|---|---|---|---|
| Tr | Transformity | $Tr = Emergy/Energy$ | Emergy of the outputs divided by energy of the products. |
| %R | Renewability | $\%R = 100 \times (R/Y)$ | Renewable inputs divided by the total Emergy of the system. |
| EYR | Emergy efficiency rate | $EYR = Y/F$ | Emergy of the output divided by Emergy of the inputs that are fed back from outside the system. |
| EIR | Emergy investment rate | $EIR = F/(R + N)$ | Emergy of purchased inputs divided by Emergy of free inputs. |
| ELR | Environmental loading rate | $ELR = (F + N)/R$ | Emergy from purchased and non-renewable natural inputs divided by Emergy from renewable resources. |
| EER | Emergy exchange rate | $EER = Y/[(\$) \times (sej/\$)]$ | The emergy exchange rate is the ratio of emergy supplied to emergy received as a product when exchanged with external systems through market acquisition. |
| ESI | Environmental sustainability index | $ESI = EYR/ELR$ | Emergy yield ratio divided by Environmental loading ratio. It expresses the yield per unit of environmental stress provided. |

(Y): Internal emergy incorporated by the system, (F): Emergy of inputs that come from the economy (N): Emergy of non-renewable natural resources, (R): Emergy of renewable resources.

*2.4. Sustainability and Aquaculture*

The concern for the environment has become necessary in the fish production process, with water conservation being one of the main aquaculture subjects studied in recent years. To maintain the legality and profitability of any aquaculture enterprise, management strategies must use mostly renewable resources, respect sustainability principles, and reduce non-renewable resources [10].

Aquaculture depends fundamentally on the ecosystems in which it operates. Therefore, it is impossible to produce without causing environmental changes [23]. However, the environment's impact can be reduced to a minimum and avoid reducing biodiversity, depletion or negative compromise of any natural resource, and significant changes in ecosystems' structure and functioning [24].

Even though aquaculture changes the natural environment, generating impacts, this concept does not refer only to the biological environment [25]. In this sense, environmental impacts are human-made activities that generate changes in the physical, biological and socio-economic environment [5].

Different aquaculture systems can generate other environmental impacts [26], and such impacts depend mainly on: the type of system (closed, semi-open, and open); the aquaculture modality (fresh or marine water); species used, and especially the density and size of production. Even so, in any production, the environmental impact occurs through three processes: the consumption of natural resources, the transformation process (processing), and the generation of final products (waste).

For aquaculture, the discussion from the 1990s on developing and adopting codes of conduct, Best Management Practices (BPM), and operating standards (among others) was fostered to mitigate its impacts [27]. For example, the objective of BPM in aquaculture is: to provide a system that reduces the negative impact on social and environmental aspects, reduces the cost of production, increases profitability, reduces waste and pollution, gains or maintains access to new markets, and in addition to promoting the regularization of aquaculture enterprises [28].

Adopting best practices in aquaculture management will indeed generate benefits for the producer. According to Valenti [29], best management practices in aquaculture can ensure the sustainability of the environment within production systems and maintain a healthy ecosystem. In this case, the authors recommend: prioritising the growth of native species, balanced use of rations and adequate food management to avoid the waste that may pollute the environment, maintenance of water quality, control of fertilisation to prevent excessive use of fertilisers, restriction of the use of chemical products, carrying out compatible sanitary management, use of the polyculture or consortium within the farms, training, and qualification of employees, among others.

Every ecosystem has a limit that guarantees its use so that it does not have negative impacts, which can be recognised as carrying capacity. For example, the carrying capacity in aquaculture would produce a certain amount of organisms, such as fish, molluscs, shrimp, or others, without significantly altering the crop's surrounding ecosystem.

One of the problems in aquaculture is eutrophication, with the accumulation of nutrients such as Phosphorus (P) and Nitrogen (N) in the water. Acting as fertilisers, P and N facilitate the proliferation of unicellular alga, changing the colour of the water, usually making it a 'green soup'. Subsequently, these algae's mortality is common, generating low oxygen dissolved concentrations in the water that promotes massive fish mortality [30]. In this way, respecting the ability to support the aquaculture industry's environment enables the ecosystem's sustainability where the activity is inserted and may avoid negative economic impacts for the aquaculture farmer [5].

From this context, it is possible to perceive the importance of creating economic conditions to make viable technologies that could reconcile animal production and environmental preservation. In this debate, energy is a central point. Through the discussions presented and based on sustainability principles, economic activities such as aquaculture, in particu-

lar, can reach sustainable production levels by providing and incorporating processes and methods that aim at more sustainable production.

## 3. Results

### 3.1. The Emergy Assessment Contributions in World Aquaculture

The emergy assessment method has been used to analyse different production systems in aquaculture scales. According to the topics studied, the subjects most covered among recent publications with the application of emergy are integrated assessments of ecosystems dominated by man [31], sustainability assessment [32], environmental impact [33], and assessment and the combination of emergy assessment with other methods [10]. The number of articles found in the search was mainly distributed in the Journal of Cleaner Production (80), Ecological Indicators (46), Ecological Engineering (31), Sustain-ability Switzerland (31), and Ecological Modelling (21) (Figure 3).

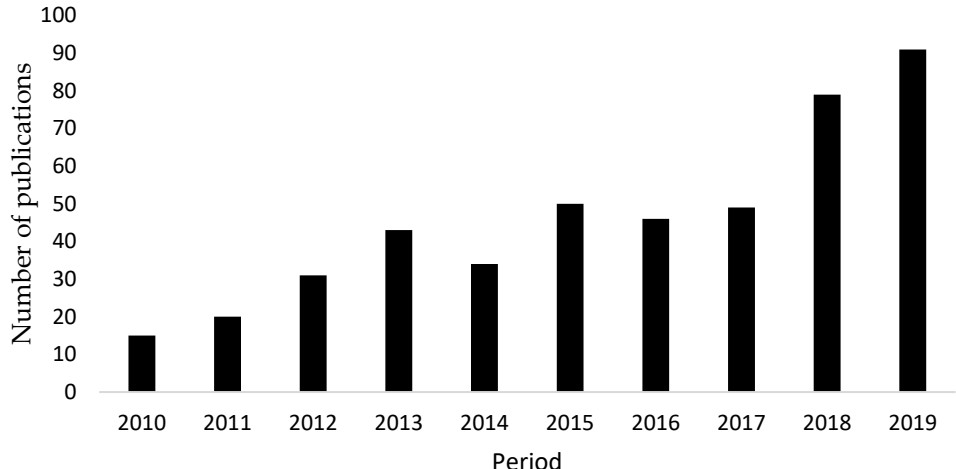

**Figure 3.** The number of publications on the research platforms (2010–2019) addressing the search terms "emergy" and "aquaculture".

The articles collected revealed the adhesion of authors and co-authors and the concern in favour of themes related to sustainability in global aquaculture. The studies also address the analysis of various production systems and sustainable development of the aquaculture sector. Of the articles collected in the selected research bases, 17 were related to the emergy used in aquaculture (Figure 4). After reading all the articles in total, it was possible to classify the countries of publication and prepare a timeline evaluating the evolution of the number of articles published on emergy in the last ten years.

This section may be divided into subheadings. It should provide a concise and precise description of the experimental results, their interpretation, and the experimental conclusions that can be drawn.

China stood out with ten articles dealing with issues involving emergy assessment methodology in aquaculture. Among the scientific journals with articles related to this theme were the Journal of Cleaner Production (4), Ecological Indicators (3), Ecological Engineering (2), Journal of Environmental Management (2), 3rd International Conference on Water Resource and Environment (1), Acta Ecologica Sinica (1), Aquaculture (1), Ifip International Federation for Information Processing (1), Journal of Fisheries of China (1), Reviews in Aquaculture (1).

After compiling the data, the network of authors and their co-authors was built (Figure 5), where we sought to discover the interfaces and relationship of works elaboration.

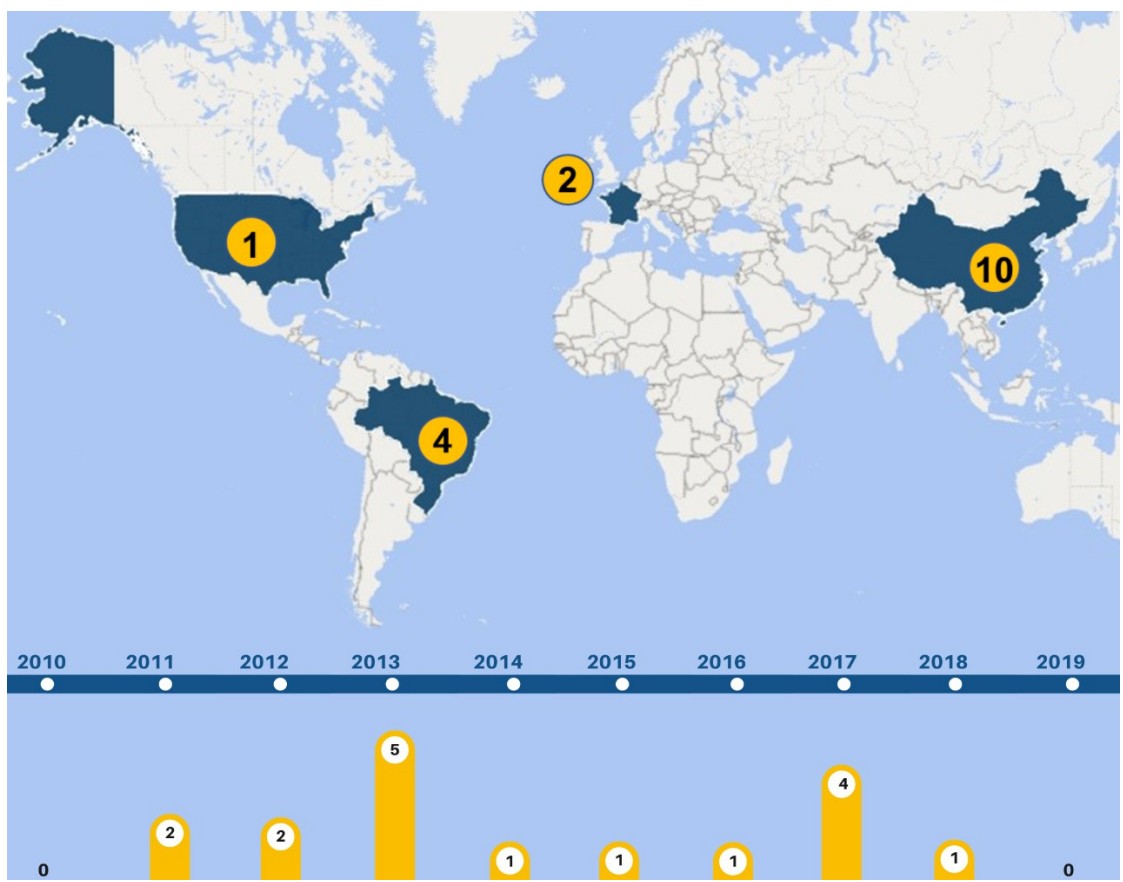

**Figure 4.** Timeline of collected articles and their countries of publication.

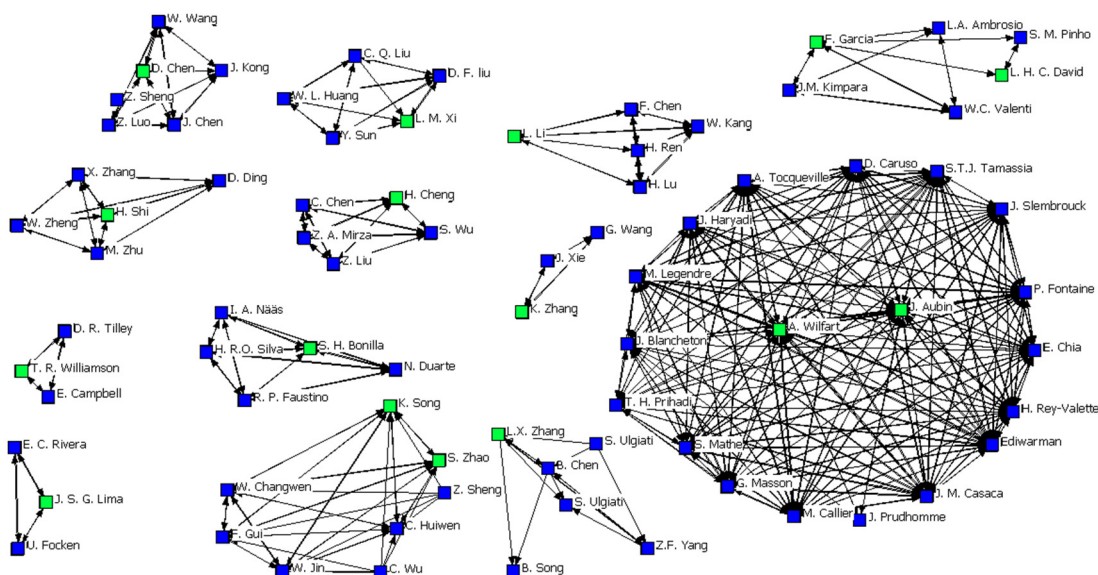

**Figure 5.** Network of authors and co-authors of articles included in the research. Source: Prepared by the author at Ucinet.

Of the 17 articles analysed, 16 articles were written by different researchers in terms of authorship (green squares) and co-authorship (blue squares). 13 different groups were formed, evidencing the lack of a relationship of publications between them. However, some authors carried out works in co-authorship linked to the same research networks.

Authors A. Wilfart and J. Aubin worked on two articles together, in 2013 and 2017, in France; and two other articles with different authors (K. Song and S. Zhao in 2013 [33]), both in China. In the articles by A. Wilfart [10] and J. Aubin [23], agroecology concepts were treated to evaluate the performance of different polyculture systems by combining emergy assessment and life cycle analysis (LCA).

The works involving the authors K. Song and S. Zhao were separate subjects. The first evaluated the production system of Large yellow croaker (*Larimichthys crocea*), and S. Zhao used the combination of two methodologies, ecological footprint and emergy, to assess the sustainability of a small fish farm in China.

The network analysis of authors and co-authors identified that the groups working with emergy are still under construction. Future interactions between these and other groups that use other tools for measuring sustainability would promote the development of emergy analysis and studies with a more systemic approach.

When analysing the frequency of terms used in the titles and keywords of the publications (Figure 6), words linked to emergy and aquaculture methodology stand out again.

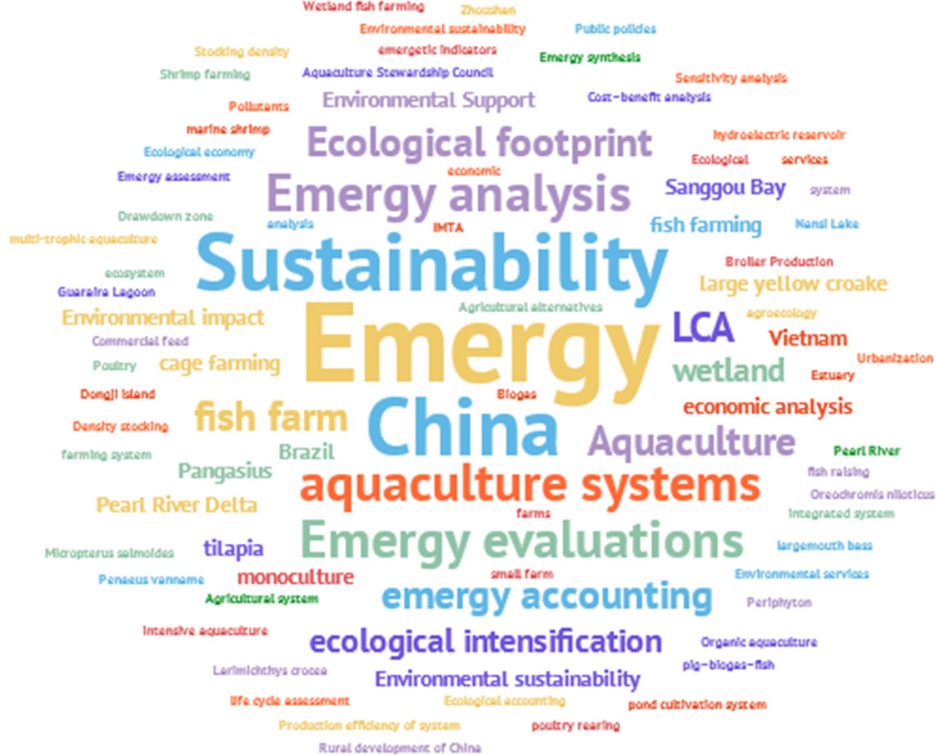

**Figure 6.** Keyword cloud used in selected articles. Source: elaborated by the author on the Infogrann platform.

It can be seen in the figure above that the keywords of the analysed articles focus on points addressed by institutional agents to aquaculture. The size of each word indicates its frequency and relevance to a given topic.

The words focused on emergy topics and associated with other sustainability analysis methods (such as Life Cycle Analysis and Ecological Footprint) were the most cited. China was the country that concentrated the major number of published studies dealing with evaluating sustainability in cage farming and semi-intensive systems in lake environments, flooded areas, and important rivers in the region.

Among the aquatic animals studied in the articles (Figure 7), most are fish of the species *Hypophthalmichthys molitrix* (8.2%) and *Oreochromis niloticus* (8.2%). On the other hand, most species are freshwater (62.63%) and carnivorous food habits (43.83%).

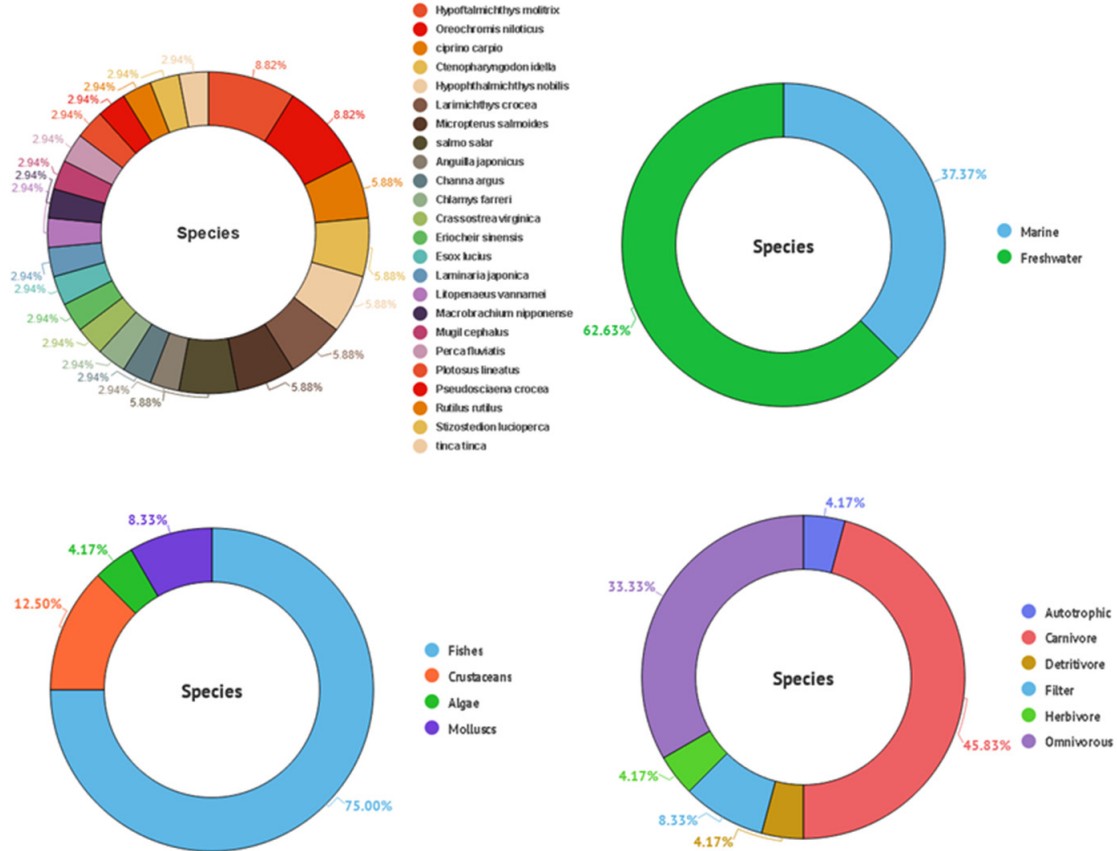

**Figure 7.** Description of aquatic organisms in articles evaluated with Emergy.

Generally, marine fish production that is directed to carnivorous species was observed. For example, the *Larimichthys crocea* and *Salmo salar* were evaluated in the studies by [33] and Aubin et al. [23], respectively. Quite different from that in freshwater fish farming, where omnivorous species dominate and expand production [34]. Although marine fish have a high production cost due to their intensive farming system (obtention of fingerlings or even in the production of live food), the production of carnivorous species (usually marine species) is more economically attractive and valuable than omnivorous species [35].

According to FAO [34], aquaculture produces mainly aquatic organisms in freshwater, and, in some countries, in-land aquaculture also uses saline and alkaline waters to cultivate species naturally adapted to environments. Introduced species, including marine species, that tolerate water conditions adequately meet fish farmers' expectations.

Carp species are the most produced in the world with 29%, followed by Tilapia with 8% of the total produced in 2016 [1]. The works presented [4,10,36,37] evaluated the sustainability of production in polyculture systems for silver carp, common carp, and grass carp.

The use of emergy assessment in aquaculture has recently become popular. The collected articles support authors and co-authors on the concern favouring themes related to sustainability in aquaculture. Seventeen articles were found in the search, and the main results and conclusions of these studies are shown in Table 3.

Table 3. Review studies that applied emergy assessment to measure aquaculture production systems' sustainability between 2010–2019.

| Fish Species | Production System | Objective | Main Results and Conclusions | Reference |
|---|---|---|---|---|
| Japanese eel (*Anguilla japonicus*), Largemouth black bass (*Micropterus salmoides*), Snakehead (*Channa argus*) and Flathead grey mullet (*Mugil cephallus*) | Monoculture and polyculture | Assessed the sustainability of three production systems through Emergy and economic assessment in China | The three systems studied showed similar emergy characteristics but different economic features. Eel farming proved the best option to improve the local economy and did not increase the environmental impact. The production of juveniles on the farm was the strategy found in all cultures to reduce production costs and the high input of resources from the economy. The study also showed that nature reserves could increase regional sustainability, although these reserves were not economically viable. According to the authors, emergy assessment has proven to complement economic assessment, production efficiency, environmental impacts, economic benefits, and ecological and sustainability of aquaculture systems. | Li et al. (2011) [31] |
| Grass carp (*Ctenopharyngodon idella*) and Silver carp (*Hypophthalmichthys molitrix*), Bighead Carp (*Hypophthalmichthys nobilis*) | Production systems: in cages with natural feed; in cages with artificial feeding; extensive feeding system with artificial feeding by grass joined around. | Compared to the different fish farming systems regarding resource use and environmental impacts, China | According to the results, the main difference between the three production systems was the emergy cost associated with feeding the fish. The emergy indicators induced that intensive production added to commercial food was not sustainable. The ESI (emergy sustainability index) is less than 0.4, while the other source systems have higher sustainable values. However, the use of plankton and grass was not economically viable. | Zhang et al. (2011) [36] |
| Grass carp (*Ctenopharyngodon idella*) and Silver carp (*Hypophthalmichthys molitrix*) | Extensive polyculture | They evaluated and compared four local production systems' environmental performance: planting maise, mushrooms, carp, and duck farming in China. | The results showed that ducks and mushrooms as diversifying production were not sustainable. On the contrary, an extensive carp polyculture system showed the best emergy performance, mainly with the indicators of renewability and sustainability. | Zhang et al. (2012) [37] |
| Shrimp (*Penaeus vannamei*) | Semi-extensive and extensive system | The sustainable performance of Brazil's conventional and organic shrimp production was evaluated and compared. | Both systems had a high emergy flow of non-renewable resources. However, the results showed that renewability, emergy production rate, and emergy investment ratio (EER) favoured shrimp's organic cultivation. New improvements in the organic system were indicated to increase efficiency and guarantee economic sustainability, given the low price practised for organic shrimp sales. The authors suggested that multitrophic systems would be beneficial because they would increase and diversify production without increasing commercial feed consumption, the main non-renewable source used in aquaculture. | Lima et al. (2012) [38] |

**Table 3.** *Cont.*

| Fish Species | Production System | Objective | Main Results and Conclusions | Reference |
|---|---|---|---|---|
| Fishes | Intensive offshore cage system | The authors sought to answer questions about using the emergy assessment methodology and ecological footprint to evaluate the aquaculture production system considering the nature of the method, data quality, and results proposed by both methods. | According to the authors, there is a need to improve the evaluated methods (emergy synthesis and ecological footprint). In addition, the input flows (data collection) must be carefully processed due to their significant impact on the results. Moreover, data that aim to carry out comparative analyses are necessary to improve these methodologies' interpretation and quality. | Chen et al. (2013) [39] |
| Large yellow croaker (*Larimichthys crocea*) | Intensive system in cages | The article sought to analyse, utilising an emergy assessment, the production system of yellow croaker, characterising the use of resources, environmental impact, and the general sustainability of the studied system. | The authors understood that the system depended more on inputs from external resources. The ESI (emergy sustainability index) and EISD (sustainable development emergy index) indices indicate that the yellow croaker production system is less sustainable. Based on sensitivity analysis, the ESI and EISD indices were high due to half the number of fry entries and doubled the number of entries in the system of chemical compounds in water. In this way, the authors suggested reducing feed inputs for better efficiency, implementing aquaculture facilities in areas with more precipitation, improving the proportion of local renewable resource inputs, and the efficiency of work or farming. | Song et al. (2013) [33] |
| Kombu (*Saccharina japonica*) and scallops (*Azumapecten farreri*) | Monoculture and polyculture | They assessed monoculture's ecological benefits of kelps and scallops and the polyculture of kelps and scallops in China. | Polyculture had the highest sustainability indicator compared to two other isolated monocultures. The study showed that integration is an alternative to a sustainable aquaculture model. | Shi et al. (2013) [40] |
| Atlantic Salmon (*Salmo salar*), Common Carp (*Cyprinus carpio*), Tench (*Tinca tinca*), Roach (*Rutilus rutilus*), European Perch (*Perca fluviatilis*), sander (*Stizostedion lucioperca*) e Northern pike (*Esox lucius*) | Intensive recirculation; Extensive polyculture; Semi-intensive polyculture. | Evaluated the environment and systems performance by combining emergy assessment and life cycle analysis in France | The recirculation system produced less environmental impact than the two polyculture farms with a low feed conversion rate. The recirculation system has been identified as highly dependent on economic resources. Polycultures incorporated renewable resources but produced more significant environmental impacts due to the inefficient use of economic inputs. The study emphasized that the factors necessary for the successful ecological intensification of fish farming should minimize economic inputs, reduce feed conversion rate and increase local renewable resources. Combining these two methods was a practical strategy to study the optimization of the efficiency of aquaculture systems. | Wilfart et al. (2013) [10] |

**Table 3.** *Cont.*

| Fish Species | Production System | Objective | Main Results and Conclusions | Reference |
|---|---|---|---|---|
| Large yellow croaker (*Pseudosciaena crocea*) | Intensive offshore cage system | Sustainability assessment using ecological footprint and emergy assessment methods on a small fish farm in China. | The "emergy footprint" was 1,953.9 hectares, 14 times greater than the carrying capacity and 293 times greater than the physical area occupied by fish farming. About 2,000 hectares of ecologically productive land were needed to support fish farming. The most usual entrances of the emergy footprint were food, fry and fuel. The authors concluded that combining these two assessment methods could be a practical and efficient means of comparing and monitoring fish farming's environmental impact. Besides, the high dependence on external contributions has affected the sustainability of fish farming. | Zhao et al. (2013) [41] |
| Nile tilapia (*Oreochromis niloticus*) | Cage farming | The sustainability of tilapia cage farming in a hydroelectric reservoir was evaluated, and management techniques and public policies contributing to this production system's sustainability were also evaluated. | The emergy evaluation showed that the production system Is Inefficient and pointed out the causes. To solve the problem was suggested to adopt measures that proportionally reduce the supply of commercial feed and increase the inflow of renewable resources. Moreover, management changes include reducing stocking density and increasing the organic load's dilution area. | Garcia et al. (2014) [42] |
| Eastern Oyster (*Crassostrea virginica*) | Floating and bottom cage system | An assessment and comparison of the different production systems' sustainability were conducted on an aquaculture farm in the United States. | The emergy results from both systems had acceptable rates referring to the economy's resources, such as human labour, purchase of juveniles, fuels, goods, and services. In addition, oyster production farms were supported by a larger percentage of local renewable resource sources than other aquaculture products, mainly by particulate organic matter and estuarine water circulation. Overall, the study showed that oyster production farms have less impact on the environment, greater sustainability, and benefit to society than other aquaculture forms. The authors suggested reducing fuel and electricity as two efficient ways to increase the sustainability of the oyster aquaculture farm. | Williamson et al. (2015) [43] |

**Table 3.** *Cont.*

| Fish Species | Production System | Objective | Main Results and Conclusions | Reference |
|---|---|---|---|---|
| Nile tilapia (*Oreochromis niloticus*), chicken of the Hubbard genetics | Greenhouse tunnel system; cage system | Using the Emergy of the role of natural services in an area of land, it evaluated the role of a sink for ammonia emissions from a poultry production shed in the region of Mato Grosso and phosphorus from an aquaculture farm in São Paulo, Brazil. | The results suggest that poultry farming seems to be a thousand times more "eco-efficient" than aquaculture, and has a smaller support area. Accounting for environmental services to dilute emissions was necessary to assess the sustainability of processes and quantify externalities properly. The challenge is to adjust human production patterns to the biosphere's ability to absorb by-products without overload. To this end, the services provided by natural capital have to be appropriately assessed and finally quantified in terms comparable to the economy. | Bonilla et al. (2016) [44] |
| Pigs and fish | Polyculture and recirculation system | This article aimed to adapt this concept of sustainability for fish farming using agroecological principles and the structure of ecosystem services. | The method was developed from published literature and applications in four study sites chosen for their differences in production intensity: polyculture ponds in France, integrated pig and pond polyculture in Brazil, striped catfish in Indonesia, an aquaculture system for salmon recirculation in France. Based on the construction of a scenario, aquaculture's ecological intensification was defined as the use of ecological processes and functions to increase productivity, strengthen ecosystem services, and decrease disservices. The expected consequences for agricultural systems include greater autonomy, efficiency, and better integration in the surrounding territories. Ecological intensification requires territorial governance and helps improve from a sustainable development perspective. | Aubin et al. (2017) [23] |
| Poultry and fish | Poultry and fish polyculture | The study evaluated and compared the environmental performance of three monocultures in China. | Polyculture produced the most significant inflow of renewable resources, showing less dependence on the economy than other crops. In addition, emergy indicators showed that the fish farming system was more sustainable when compared to others. The authors recommended public policies that encourage sustainable agricultural production by local producers and the use of clean energy. | Cheng et al. (2017) [45] |

**Table 3.** *Cont.*

| Fish Species | Production System | Objective | Main Results and Conclusions | Reference |
|---|---|---|---|---|
| Water chestnut (*Trapa bispinosa*), Silver carp (*Hypophthalmichthys molitrix*), Bighead carp (*Aristichthys nobilis*), snail (*Cipangopaludina cathayensis*), Chinese mitten crab (*Eriocheir sinensis*), shrimp (*Macrobrachium nipponense*), snail (*Radix auricularia*), Common carp (*Cyprinus carpio*) | Polyculture system | This study, compared the eco-economic systems under different polyculture models between China's Xiaoxidian and Dujiadiana areas. | Based on the results, the authors could observe that the Xiaoxidian ecological system has higher emergy production and economic income per unit area than the Dujiadian area. In comparison, the Dujiadiana area has a higher emergy production rate and a lower environmental load rate. Therefore, the Dujiadian area is less sustainable due to humans' constant overload of non-renewable energy. Therefore, adjusting and optimising the aquaculture system's management in the Xiaoxidian area was recommended to find a stable balance between environmental sustainability and economic benefits. | Xi et al. (2017) [4] |
| Largemouth black bass (*Micropterus salmoides*) | Semi-intensive system. | In this study, the objective was to evaluate the benefits and driving forces of the M. salmoides aquaculture system using an emergy analysis method from the ecological and economic points of view (country) | The lower ESI (Emergy Sustainability Index) with EISD (Emergy Sustainable Development Index) and the higher ELR (Environmental Loading Rate) showed that emerging inputs from acquired external resources achieved a more significant effect than Emergy from renewable environmental resources in the aquaculture system of M. salmoides. The system was more dependent on emergy inputs from externally acquired resources, which indicated that the production of M. salmoides is less sustainable. The result showed that measures that reduced feed inputs improved their use as the use of feed and additives with a low feed coefficient could reduce the inputs of acquired external resources and then raise the ESI and EISD of the feed system. Aquaculture of M. salmoides. Integrated aquaculture was another method that could achieve the same result. | Zhang et al. (2017) [46] |
| Nile tilapia (*Oreochromis niloticus*) | Cage system | The study's objectives were to identify the contributions of nature and the economy to raising Tilapia in cages using Emergy to assess whether using the periphyton as a complementary food and whether reducing storage density could improve the system's sustainability of production. (parents) | Three different production systems were evaluated and compared: using traditional stocking density adopted by farmers (80 kg/m$^3$) with 100% of the recommended daily ratio and without substrates for the periphyton (TRAD); traditional stocking density (80 kg/m3) with 50% of the recommended daily ration and with substrates for periphyton (TDS); lower density (40 kg/m$^3$) with 50% of the recommended daily food and with substrates for the periphyton (LDS). Based on the results' interpretation, the authors concluded that tilapia production in cages is highly dependent on economic resources, and animal feed is responsible for this. Therefore, from the emergy study, it was possible to identify that using periphyton to feed fish in cultivation combined with a reduction in artificial feeding and the use and reduction of stocking density should be encouraged to promote tilapia sustainability. | David et al. (2018) [32] |

Emergy assessment has been used to assess and compare sustainability in various aquaculture systems such as monocultures and polycultures production systems (intensive, semi-intensive, extensive) and more traditional alternatives according to the region studied. These analyses (Table 3) were made in production at different scales, species, locations, levels of intensification, and structures.

However, when transformed into the same unit (emjoule), all these particularities can be compared even in different scenarios. As the model permits, each productive system had its sustainability evaluated within the environmental, economic, and social context inserted, representing a detailed view of all processes and can indicate where are the solutions or problems in each system.

In the study by Zhang [36], the authors demonstrated that applying emergy assessment allowed identifying the emergy of each item or input needed for production, making it possible to modify different production flows. Furthermore, the analysis showed that more sustainable management and actions could benefit the environment and the local economy.

One of the main input items in the evaluated aquaculture systems is commercial feed (31.99%) as the primary source of emergy expenditure in intensive or semi-intensive systems, followed by the purchase of juveniles and other inputs such as pesticides, machinery, oil, and services contributed with 88.17% in the production of cage fish and 73.30% in the extensive semi-natural fish farming system of the different species cultivated.

Studies such as those by Wilfart [10], Garcia [42], and David [32] pointed to artificial food as the primary input for the instability of the production system, representing an average of 65.00; 76.43 and 67.08%, respectively, suggesting the reduction of artificial foods and the increase of natural foods or alternating both as an alternative for more sustainable production. Besides, changes in the production systems schedules that aim to carry out the rearing and fattening phases on the same farm are also encouraged. As these measures mentioned have low emergy, they could increase the renewability of the systems since it would reduce the external inputs that increase the cost of production.

From the interpretation of the synthesised results (Table 3), it is clear that aquaculture systems are not sustainable as production intensifies. According to FAO [1], this occurs in intensive monocultures in small spaces that seek to serve the avid world market in short periods. These fish production systems with high dependence on economic resources have a high impact and are not sustainable. The intensive cage farming of yellow Corvina is not sustainable, according to Song et al. [33], with registered values of transformity (Tr) of 1.46E+06 se/J, sustainability index (ESI) of 0.011 and a high environmental loading rate (ELR) 91.10, well above those found in systems in integrated "pig-biogas-fish" production systems evaluated by de Wu [21], with sustainability index (ESI) values of 1.17 and high environmental loading rate (ELR) 0.90.

Natural foods were cited as an alternative in other cultivation systems evaluated by Zhang [36], comparing production in different systems. They showed that sustainability was higher in extensive ESI systems (4.61) and lower ELR indices (0.38) when compared to an intensive ESI (0.38) and ELR (2.73) system, respectively. In the study by David [32], the authors evaluated tilapia production in cage farming and verified that the use of periphyton as food combined with a reduction in commercial feed is an effective alternative for the production of the species in a more sustainable way, with better ESI (0.35) and ELR (3.63) values compared to the system with a high density of fish and without the use of periphyton as food ESI (0.17) and ELR (6.81).

Suppose current aquaculture demands are considered, such as the scarcity of natural resources (marine fish meal and fish oil) and the growing demand for more sustainable food. In that case, the trend is to search for systems and management that meet market demand and respect the laws and environmental conditions. According to the studies by Zhao [41], Aubin [23], and Xi [4], the reduced use of renewable resources and high consumption of resources in the economy contribute to less system stability since when used sustainably; renewable resources make the system eco-nomically less dependent and more balanced.

Concerning the alternatives to optimise the use of resources, reducing the dependence on economic inputs, mainly commercial feed, the use of polyculture or integrated aquaculture systems were the most indicated by Lima [38], Wilfart [10], Shi [40], and Cheng [41] since the efficient use of different trophic levels reduces the production costs and the emission of pollutants to the environment. Furthermore, the literature proposes creating or adapting public policies that encourage farmers to adopt sustainable practices on their properties [37].

The elaboration of regulations that consider the carrying capacity of the systems and the use of natural resources must be taken into account. Analysing Zhang [36,37], Song [33], and David [32] (Table 3), it was possible to observe that the evaluated products concerning emergy exchange (EER) cost less than they should if the environmental value were considered. It shows that in a less intensive system, more resources are used and, therefore, more free resources he delivered to the buyer.

In turn, if the same system buys goods and commodities for its operation, the environmental resources incorporated in these purchases are generally much smaller, depending on the level of development of the surrounding economy [36].

### 3.2. Water as a Relevant Flux in the Emergy Analysis

As aquaculture's primary resource, water needs to be viewed with more care in emergy synthesis. David et al. [47] already stressed three main issues when applying emergy analysis on aquaculture: classification of water as a renewable or non-renewable resource, outdated unit emergy values, and how water is accounted for in emergy tables.

Generally, emergy flows are classified as renewable or non-renewable natural resources depending on their characteristics [48]. The second could be updated by international institutions (such as the International Society for the Advancement of Emergy Research -ISAER). Here we will not discuss these two first, as they are "easy" to adjust and, if so, could be used in any production system. Our main subject is the third one, which depends on the production system's characteristics.

It is easy to identify that water accountability lacks criteria, as shown in Table 4. In the articles accessed, one of the major problems was not the classification between renewable and non-renewable, but whether water emergy is considered (almost 30% do not consider water in their accountings). Another point to be stressed is that even classified as renewable or non-renewable, the UEV variation reached 1.0 E+13 in both classes. No study specifies precisely how they calculate the water emergy nether if they consider the water that passes through the aquaculture system.

**Table 4.** Water classification and unit emergy values (UEVs) considered in emergy studies applied to aquaculture systems.

| Reference | Water Source | Classification | UEV (sej/J) |
|---|---|---|---|
| Li et al. (2011) [31] | River water | Renewable | $5.01 \times 10^4$ |
| Zhang et al. (2011) [36] | Groundwater | Non-renewable | $8.06 \times 10^4$ |
| Zhang et al. (2012) [37] | Groundwater | Non-renewable | $8.06 \times 10^4$ |
| Lima et al. (2012) [38] | Lake water | Non-renewable | $3.71 \times 10^{11}$ |
| Chen et al. (2013) [39] | | Not Considered | |
| Song et al. (2013) [33] | River water | Renewable | $1.46 \times 10^{13}$ |
| Shi et al. (2013) [40] | Rainwater | Renewable | $6.65 \times 10^{12}$ |
| Wilfart et al. (2013) [10] | Groundwater | Non- renewable | $1.1 \times 10^{17}$ |
| Zhao et al. (2013) [41] | | Not Considered | |
| Garcia et al. (2014) [42] | Spring water | Renewable | $1.66 \times 10^5$ |
| Williamson et al. (2015) [43] | | Not Considered | |
| Bonilla et al. (2016) [44] | | Not Considered | |
| Aubin et al. (2017) [23] | | Not Considered | |
| Cheng et al. (2017) [45] | Rain chemical energy | Renewable | $1.81 \times 10^{16}$ |
| Xi et al. (2017) [4] | River/Rain chemical energy | Renewable | $6.4707 \times 10^{17}$ |
| Zhang et al. (2017) [46] | River water | Renewable | $9.77 \times 10^{15}$ |
| David et al. (2018) [32] | Rainwater | Renewable | $2.36 \times 10^4$ |

Although the international community concept of "water as an economic good" is generally recognised, disagreement about how to estimate the value of water is an actual concern [49]. Nowadays, water evaluation remains a very elusive subject and needs a unifying approach to solve this problem [50]. Furthermore, the increase in water scarcity is pressing for its pricing, showing that it will be an essential method for water resource management and sustainable use [49], including utilising emergy analysis as a proxy in the aquaculture industry [32].

First of all, the main concern is quantifying water use. The general hydrological equation (inflow = outflow $\pm$ change in storage) can be used for water use by ponds for inland aquaculture projects. The possible inflows to aquaculture ponds are precipitation, storm runoff, stream inflow, groundwater seepage, and regulated inflow; the possible outflows are evaporation, transpiration, seepage, overflow, regulated discharge, and consumptive use [51]. Pond aquaculture is more water-intensive than most other food production methods, even with water conservation measures. For example, in the south-eastern USA, the water requirements for irrigation of usual crops are $30 \pm 40$ cm year $\pm 1$ and rice, a more water-intensive crop, requires the application of $60 \pm 80$ cm year $\pm 1$ [52], less than the requirements for channel catfish farming that uses $50 \pm 100$ cm year $\pm 1$ when ponds are not drained annually and $200 \pm 250$ cm year $\pm 1$ when they are drained [53]. Nevertheless, aquaculture water is evaluated or quantified incorrectly for emergy analysis by considering the total volume that flows through the system and not the water used [47].

Secondly, it is crucial to maintain stable water quality, mainly for increased aquaculture production. It is well known that the impact of dissolved nutrient loading from marine fish farms around the world directly impacts water quality and secondary impacts on primary production, besides the formation of harmful algal blooms [54]. With cage aquaculture, nutrients are directly discharged into the environment [55]. On the other hand, aquacultural water is often polluted by microorganisms or chemicals from domestic and industrial waste. Water quality guidelines for aquaculture purposes usually apply to BOD, COD, T-P (total phosphorous), T-N (total nitrogen), DO, SS (suspended solids), and coliforms in rivers, lakes, and seawater [56].

In aquaculture, the loading of nutrients is defined as the difference between its entrance by fertilizers and feed. It is harvested from finfish, crustaceans, molluscs, and seaweeds. On average, the production of finfish and crustaceans results in a net nutrient loading, while the production of molluscs and seaweeds is negative [55]. This loading must be computed in the emergy analysis as ecosystem harm. According to Costa-Pierce [57] summarized critical resource use in aquaculture, and trends from 2000 to 2050, from that, could be highlighted as severe water competition growing with alternative users, freshwater use conflicts; droughts increase in aquaculture production zones closing many pond areas; and rapid decrease in the costs and increased efficiencies of intensive recirculating systems. They all put more pressure to show the correct use of this resource by aquaculture.

Concerning emergy, the water value is provided by both the chemical potential energy (a crucial source of biological production influenced by water sediment) and the geopotential energy (the work that water running off the landscape can do as it falls from higher elevation to lower elevations) [58,59]. So, volume, quality, and hydraulic potential are the most important factors to measure emergy in water. Since water is the most used resource in aquaculture, wrong interpretations of water evaluation, water classification, and outdated unit emergy values would result in high inaccuracies in the final numbers and lead to wrong interpretations [47].

As all these aspects of water are important, the maintenance and the change aquaculture can make to them are relevant to the environment and other aquaculturists close or downstream. However, these characteristics depend on which system will be used. The three systems identified in this study were earth pond, RAS, and cage culture (Figure 8). For each one, the water is treated differently, which is essential to establish the main aspects of water quantity, quality, and hydraulic potential. In these systems, water flows into the system to fill the ponds and/or cages where aquaculture production occurs. The volume

of water entering is the same as that leaving (except for RAS), but the latter has a lower quality with higher concentrations of nutrients and organic compounds. As earth pond and cage culture systems rarely have a water treatment process unit, this low-quality water is discharged directly into natural water bodies, potentially causing a disservice to the environment and society [47].

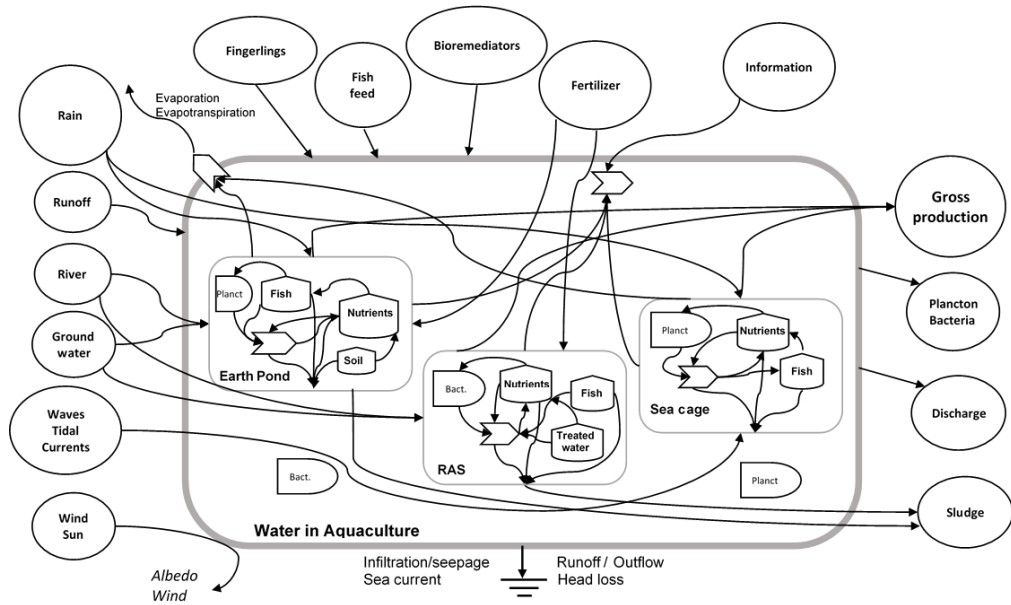

**Figure 8.** Diagram of water energy systems in aquaculture of three systems identified in this study land pond, RAS, and cage culture.

For inland water aquaculture quantification, stream inflow, runoff, transpiration, and consumptive use are seldom major factors in water use, with an important exception when de pond water is used as an irrigation source [51]. In this same way, aquaculture water quality must address dissolved oxygen and nutrients (N and P mostly), phytoplankton, organic matter (waste), head loss from surface water that is not pumped, and recharged loss if water is pumped from wells. In resume, it is necessary to identify the system's outputs concerned with water flow in the production system, as is proposed in Table 5.

**Table 5.** Usual importance of water quantity/quality in aquaculture systems (earth pond, recirculating aquaculture system, and sea cage) to calculate water emergy synthesis.

| PARAMETER | EARTH POND | RAS | SEA CAGE |
|---|---|---|---|
| **INPUTS** | | | |
| Rain:<br>• Geopotential energy<br>• Chemical energy<br>• Nitrogen<br>• Phosphorous | Important | Not important<br>(rain can be collected<br>to be used) | Low important |
| Spring/well/creek/stream/river/lake inflow:<br>• Geopotential energy<br>• Chemical energy<br>• Organic matter<br>• Nitrogen<br>• Phosphorous<br>• Phytoplankton<br>• Sediment | Important | Very important | Not applicable |

**Table 5.** *Cont.*

| PARAMETER | EARTH POND | RAS | SEA CAGE |
|---|---|---|---|
| Superficial Run-in/Runoff:<br>• Chemical energy<br>• Organic matter<br>• Nitrogen<br>• Phosphorous<br>• Sediment | Important | Not applicable | Not important if offshore |
| Groundwater inflow:<br>• Geopotential energy (gushing well)<br>• Chemical energy<br>• Organic matter<br>• Nitrogen<br>• Phosphorous<br>• Sediment | Important | Very important | Not applicable |
| Waves | Not applicable | Not applicable | Very important if offshore |
| Tidal | Not applicable | Not applicable | Very important if inshore |
| Sea currents:<br>• Geopotential energy<br>• Chemical energy<br>• Organic matter<br>• Nitrogen<br>• Phosphorous<br>• Phytoplankton<br>• Sediment | Not applicable | Not applicable | Very important |
| **OUTPUTS** | | | |
| Evaporation | Very important | Low important | Low important |
| Evapotranspiration | Important | Not applicable | Not applicable |
| Infiltration/seepage/soil moisture:<br>• Chemical energy<br>• Nitrogen<br>• Phosphorous | Important (depends on soil characteristics and compactness) | Not applicable | Not applicable |
| Runoff:<br>• Geopotential energy<br>• Chemical energy<br>• Organic matter<br>• Nitrogen<br>• Phosphorous<br>• Phytoplankton<br>• Sediment<br>• Topsoil loss | Important (depends on micro drainage structure) | Not applicable | Not applicable |
| Spillway outflow:<br>• Geopotential energy<br>• Chemical energy<br>• Organic matter<br>• Nitrogen<br>• Phosphorous<br>• Phytoplankton<br>• Sediment | Very important | Not important (unusual situation) | Not applicable |

**Table 5.** *Cont.*

| PARAMETER | EARTH POND | RAS | SEA CAGE |
|---|---|---|---|
| Regulated discharge:<br>• Geopotential energy<br>• Chemical energy<br>• Organic matter<br>• Nitrogen<br>• Phosphorous<br>• Phytoplankton<br>• Sediment | Very important | Very important | Not applicable |
| Sea currents:<br>• Geopotential energy<br>• Chemical energy<br>• Organic matter<br>• Nitrogen<br>• Phosphorous<br>• Phytoplankton<br>• Sediment | Not applicable | Not applicable | Very important |
| Gross production (fish/seafood):<br>• Humidity<br>• Organic matter?<br>• Nitrogen<br>• Phosphorous | Very important | Very important | Very important |

Obs.: 1-Wind influences the gas changes in the water but will not be considered here because it is measured in primary fluxes of renewable sources of Emergy. 2-In some specific cases, these impact levels can change. Each sub-item will need a previous analysis to insert in the emergy account as sometimes the influence could be negligible or absent. 3-The emergy synthesis items were compiled by Brown and Bardi [60] and Fang [61].

Vallee [62] defines renewable water resources as "the average annual flow of rivers and recharge of aquifers generated from endogenous precipitation" to identify renewable resources from those that are not. While water undoubtedly undergoes a natural cleaning process and maintains a dynamic balance over time thanks to the hydrological process, human intervention has resulted in changes in the quantity of surface and groundwater reserves and the quality of the resource. Especially in the case of deep aquifers, water bodies with recharge rates ranging from hundreds to thousands of years have been used [63,64]. Therefore, regarding the recharge rate, the non-renewability of water resources is primarily associated with groundwater [65].

Besides the renewable/non-renewable discussion, the water resources consumed by agricultural production (e.g., aquaculture) can be divided into the blue and green water. The first refers to surface water and groundwater, and the second refers to the part stored in the soil that does not form runoff or leakage (derived from precipitation) and is eventually consumed by transpiration or evaporation [66]. It is another way to understand that the water's origin gives them distinct qualifications. The emergy analysis of blue and green water separately provides information for a more precise and efficient water allocation [67].

The emergy corresponding to the surface water resources of a basin is variable spatially and temporally, partly due to the differences in water quality found along its rivers. For example, low values for total dissolved solids can be observed in the upper sections, while high values can be found in the lower sections due to sediment transport or discharge from anthropogenic activities [65]. Similarly, at a single point along the river, the water quality changes over a year according to the volume of water flowing through its cross-section [68] and the erosion by runoff [68].

For calculating the water quality, methods with models based on water quality and quantity of transboundary sections (in and out of the system) are needed because they reflect the property rights of water resources [69]. In addition, ecosystem service value assessment is an essential reference for formulating ecological compensation standards in

water-receiving areas [70], as it could be used in aquaculture farms that promote a better quality of disposed water [47].

At the end of the system, the output flow is generally not considered in the emergy synthesis of aquaculture systems and, when considered, is quantified using economic approaches and accounted for as a service [47].

## 4. Conclusions

The set of information provided by the evaluation of the emergent synthesis in different articles that deal with aquaculture offers technical and scientific data that can contribute to the planning and adoption of more sustainable production systems and help ensure the long-term success of the activity.

Most studies evaluate production systems at the intermediate stage of the aquaculture chain, such as fattening systems, which are more focused on evaluating fish farming. Based on this finding, it is possible to suggest using the emergy synthesis in the other links of the aquaculture production chain to add information about energy expenditure. It is especially true in inputs such as seeds and commercial feed, which contribute significantly to the calculation of the emergy in aquaculture systems.

In addition, using emergy to evaluate the sustainability of aquaculture can promote the need to increase the use of renewable resources, using an integrated production system and other feed alternatives such as multi-trophic systems. These strategies can be part of the sustainability guidelines in fish production, promoting the welfare of the community, environment, and local economy.

As for natural resources, the emergent information identified the primary inputs of nature that directly influence the production system, among them water. This input is the central natural resource used in aquaculture systems and one of the least considered in emergy calculations. About 30% of the studies did not account for water in their emergence synthesis, and 100% did not consider it as an output product, only the core product (fish), showing the importance of more detailed analysis considering the use, impact, and emergy flow of water. Finally, a theoretical structure was proposed with the indication of the main emergent flows, which seek to clarify the procedures that aim for a correct approach to this resource. This framework promotes a better understanding of the conservation and maintenance of aquaculture activity over the years, which has in water its most significant wealth.

**Author Contributions:** Conceptualization, Ú.d.S.M., M.A.R. and D.P.S.J.; methodology, Ú.d.S.M. and M.A.R.; formal analysis, Ú.d.S.M.; investigation, Ú.d.S.M. and D.C.F.; resources, D.P.S.J.; data curation, Ú.d.S.M.; writing—original draft preparation, Ú.d.S.M. and M.A.R.; writing—review and editing, Ú.d.S.M., M.A.R., D.C.F. and D.P.S.J.; visualization, Ú.d.S.M. and M.A.R.; supervision, D.C.F. and D.P.S.J.; project administration, D.P.S.J.; funding acquisition, D.P.S.J. All authors have read and agreed to the published version of the manuscript.

**Funding:** This research was funded by National Council for Scientific and Technological Development (CNPq) (141617/2018-7), coordination for the Improvement of Higher Education Personnel (CAPES) [grant number 23038.011255/2021-17] and the Federal University of Rio Grande do Sul (UFGRS).

**Acknowledgments:** The National Council for Scientific and Technological Development (CNPq), Coordination for the Improvement of Higher Education Personnel (CAPES) and the Federal University of Rio Grande do Sul (UFGRS).

**Conflicts of Interest:** The authors declare no conflict of interest.

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
