# Peer review of "Aquaculture Sustainability Assessed by Emergy Synthesis: The Importance of Water Accounting"

_agriculture, doi:10.3390/agriculture12111947_

Round 1

Reviewer 1 Report

Dear Colleagues,

The paper needs some corrections. I put my suggestions in text.

For your future work, for scientific/common names of species please verify the international bases like: https://www.fishbase.se/

https://www.marinespecies.org/

https://www.uniprot.org/

Success!

Author Response

Dear Reviewer,
The Recommended corrections were strictly followed and highlighted in the text as requested by the journal.
Point 1: Minor punctuation errors were readjusted and the main conclusions were added to the summary.
Point 2: the words suggested in the text were adjusted and the scientific names in Table 3 updated

For more details,  see the attachment.

Reviewer 2 Report

This review study addresses the emergy approach to evaluate aquaculture ssutainability with parameters such as energy, water, or economic. A few concerns for this study are as follows,

1. For the systemic review of the 17 papers, what was the method used for systemic review? Was it meta-analysis or word mining?

2. This review found that 30% of the papers did not consider water in the emergy consideration and 100% did not consider it as a major output. Is there any reason for this since water consumption should be a major cost when sustaining aquaculture farms or  artificial ponds?

3. Could you explain what would be the purposes to analyze the networks of authors or co-authors? 

Author Response

Dear Reviewer,

The Recommended corrections were strictly followed and highlighted in the text as requested by the journal and questions answered.

For more details, please see the attachment.
